# PD-L1 Gene Polymorphisms rs822336 G>C and rs822337 T>A: Promising Prognostic Markers in Triple Negative Breast Cancer Patients

**DOI:** 10.3390/medicina58101399

**Published:** 2022-10-06

**Authors:** Andreas-Evangelos Makrantonakis, Eleni Zografos, Maria Gazouli, Konstantinos Dimitrakakis, Konstantinos G. Toutouzas, Constantinos G. Zografos, Despoina Kalapanida, Andriani Tsiakou, George Samelis, Flora Zagouri

**Affiliations:** 1Second Department of Medical Oncology, Theagenion Cancer Hospital, 546 44 Thessaloniki, Greece; 2Department of Basic Medical Sciences, Laboratory of Biology, School of Medicine, National and Kapodistrian University of Athens, 115 27 Athens, Greece; 3Department of Obstetrics and Gynaecology, Alexandra Hospital, Medical School, National and Kapodistrian University of Athens, 115 28 Athens, Greece; 41st Propaedeutic Surgical Department, Hippokrateio Hospital, National and Kapodistrian University of Athens, 115 27 Athens, Greece; 5Department of Clinical Therapeutics, Alexandra Hospital, Medical School, National and Kapodistrian University of Athens, 115 28 Athens, Greece; 6First Department of Dermatology, Syggros Hospital, School of Medicine, National and Kapodistrian University of Athens, 161 21 Athens, Greece; 7Department of Oncology, Hippocrateion Hospital, National and Kapodistrian University of Athens, 115 27 Athens, Greece

**Keywords:** breast cancer, breast cancer prognosis, immune checkpoints, molecular tumor markers, PD-L1, polymorphisms

## Abstract

*Background and Objectives*: Triple-negative breast cancer (TNBC) is a highly heterogeneous subtype that is associated with unresponsiveness to therapy and hence with high mortality rates. In this study we aimed to investigate the prognostic role of the rs822336 G>C and rs822337 T>A polymorphisms of the PD-L1 (Programmed Death-Ligand 1) in TNBC patients. *Materials and methods:* Formalin-fixed paraffin-embedded tissues from 114 TNBC patients and blood samples from 124 healthy donors were genotyped, and subsequently extensive statistical analysis was performed in order to investigate the clinical value of these polymorphism in TNBC. *Results:* Regarding rs822336 G>C, we found that the CG genotype was the most common among women that harbored Stage IV breast tumors (81.8%; *p* = 0.022), recurred (38.9%; *p* = 0.02) and died (66.7%; *p* = 0.04). Similarly, the rs822337 T>A genotype AA is associated with worse prognosis, since it was the most common genotype among stage IV tumors (72.7%; *p* = 0.04) and in TNBC patients that relapsed (75%; *p* = 0.021) and died (81.5%; *p* = 0.004). Our statistical analysis revealed that the rs822336 G>C genotype CG and the rs822337 T>A allele AA are strongly associated with inferior DFS and OS intervals. Moreover, it was revealed that women harboring mutated genotypes of both SNPs had shorter disease-free (Kaplan–Meier; *p* = 0.037, Cox analysis; *p* = 0.04) and overall (Kaplan–Meier; *p* = 0.025, Cox analysis; *p* = 0.03) survival compared to patients having normal genotype of at least one SNP. Multivariate analysis also showed that the presence of mutated genotypes of both SNPs is a strong and independent marker for predicting shorter DFS (*p* = 0.02) and OS (*p* = 0.008). *Conclusion:* Our study revealed that PD-L1 rs822336 G>C and rs822337 T>A polymorphisms were differentially expressed in our cohort of TNBC patients, and that this distribution was associated with markers of unfavorable prognosis and worse survival.

## 1. Introduction

The strong interaction between the immune system and malignant transformation is a recently recognized aspect of cancer pathogenesis that opened new horizons in cancer patients’ management, since new opportunities have arisen regarding the recognition of novel immuno-oncological biomarkers and therapeutic targets. Already, many immunological concepts have been successfully translated into novel clinical treatment strategies that have radically changed the therapeutic landscape of many cancer types, including breast cancer. For example, the recent FDA approvals of the monoclonal antibody pembrolizumab and the antibody-drug conjugate acituzumab govitecan for the treatment of triple negative breast cancer (TNBC), were considered important milestones in the management of these patients [1,2].

TNBC is the most aggressive breast cancer subtype and often is associated with unresponsiveness to therapy and hence with high rates of mortality [3]. It is now accepted that the heterogeneity of TNBC renders the current therapeutic approaches inadequate as they are unable to target the different molecular pathways that are implicated in TNBC pathogenesis. Mounting evidence suggests that different degrees of immunogenic activity contribute to the phenotypic and clinical heterogeneity of TNBC [4,5,6]. This subtype exhibits the highest tumor immunogenicity of all breast cancer subtypes [7,8,9,10,11] and nowadays ongoing research efforts are focused on the thorough characterization of TNBC immune-associated features so that immune signatures can be incorporated into the established molecular subtyping, refining thus the routine prognostic and therapeutic-decision approaches. Hereto, several independent groups have studied the immune-landscape of TNBC along with the expression of immune-related genes and the results support the notion that TNBC can be subclassified in distinct subtypes according to their immunogenomic profile. This separation of strongly immunogenic tumors from the weakly ones hold promises for a more personalized prognosis and treatment in terms of immunotherapy [12,13,14,15,16,17].

Programmed death ligand-1 (PD-L1) is one of the molecules that is steadily included in the majority of these studies. The PD-L1 is an immune checkpoint molecule that is at the forefront of breast cancer research since it seems that not only it contributes to breast neoplastic transformation, but is also a clinically useful biomarker. More specifically PD-L1 expression influences TNBC prognosis [18,19,20,21], a potential that was clearly shown by the accelerated FDA approval of the immune checkpoint inhibitor atezolizumab, despite its 2021 withdrawal due to the failure of a subsequent clinical trial [22]. PD-L1 is encoded by the *CD274* gene that is located in chromosome 9p24.1 [23]. The advent of high-throughput sequencing technologies facilitated the recognition of several single nucleotide polymorphisms in *PD-L1* and paved the way for the identification of potential, novel cancer biomarkers. A recent study suggested that the SNPs rs4143815 and rs2282055 may serve as useful biomarkers for the efficacy of nivolumab in NSCLC patients [24]. Moreover it has been found that certain SNPs of *PD-L1* are associated with NSCLC outcome [24], whereas other data suggest that some SNPs of *PD-L1* have clinical value in gastric adenocarcinoma [25] and NSCLC [26].

By the virtue of (1) the significant role of PD-L1 in prognosis and in guiding optimal treatment decisions of TNBC patients and (2) the fruitful results regarding the potential value of SNPs of *PD-L1* as prognostic and predictive markers, we aimed to examine the presence of the rs822336 G>C and rs822337 T>A in 114 FFPE (formalin-fixed paraffin-embedded) tissues obtained from TNBC patients and to correlate the SNP status with established clinicopathological parameters and the survival of the patients.

## 2. Materials and Methods

### 2.1. Clinical Characteristics of the Patients

The study cohort consisted of 114 FFPEs obtained from women with TNBC patients as well as of 124 blood samples from healthy donors. Detailed medical history, demographic data, clinicopathologic characteristics and follow-up survival information were collected for each patient, for statistical analysis (Table 1).

The current study was designed according to the most recent guidelines for reporting tumor biomarkers and was approved by the ethical committee of the “Hippokration”, University Hospital of Athens. Informed consent was obtained from all study participants. Moreover, research procedures of the study comply with the ethical standards of the World Medical Association Declaration of Helsinki.

### 2.2. Genotyping

DNA from paraffin-embedded breast tissues of patients and DNA from the blood of healthy controls was extracted from samples using the commercial Nucleopsin Tissue kit (Macherey-Nagel, Germany) according to the manufacturer’s instructions. Genotyping was performed using allele-specific PCR (polymerase chain reaction). We used the primers (5′-ACTCTCAGTCATGCAGAAAAC-3′ and 5′-ACTCTCAGTCATGCAGAAAAG-3′) with the last nucleotide complementary to the allelic variant substitution base on the point mutation in question of the gene, and a common primer (5′-AAGATGGAGTCAAACAGGG-3′). The amplified PCR products of 239 bp were then digested using restriction enzymes and analyzed by 2% agarose gel electrophoresis in the presence of FastGene 100 bp DNA ladder (NIPPON Genetics Europe, 52349 Düren, Germany) using ethidium bromide staining and ultraviolet visualization.

### 2.3. Statistical Analysis

Genotype frequencies were analyzed with the x^2^ test with Yate’s correction using S-Plus (v.6.2 Insightful, Seattle, WA, USA) software. Odds ratios (ORs) and 95% confidence intervals (95% CIs) were calculated using GraphPad (version 300, GraphPad Software, San Diego, CA, USA). All *p* values were two-sided and *p* values < 0.05 were considered significant. The survival curves were constructed using the Kaplan-Meier method and comparison of two survival curves was performed using the log-rank test. The influence of each variable on survival was analyzed by the multivariate analysis of Cox proportional hazard model. The comparisons were performed using GraphPad version 3.00 (GraphPad Software Inc., San Diego, CA, USA).

## 3. Results

### 3.1. Differential Distribution of rs822336 G>C and rs822337 Τ>A between TNBC Patients and Healthy Controls

Initially, we compared the distribution of rs822336 G>C and rs822337 Τ>A genotypes in 114 TNBC patients and 124 healthy controls. Regarding rs822336 G>C, it was found that the GG genotype was the most common both in patients (48.2%) and in healthy controls (55.6%). In the patients’ cohort, the second most abundant genotype was the CG (44.7%), whereas the CC genotype was the less common (7%). The same pattern was observed in the group of healthy donors and the corresponding percentages for the CG and CC genotype were 34.7% and 9.7%, respectively (Figure 1A). Likewise, the distribution of rs822337 T>A genotypes among patients and healthy controls was similar. Briefly, the most abundant genotype in both groups was the AA (patients: 53.5%, controls: 46%), following the AT (patients: 36.8%, controls: 41.1%) and the TT genotypes (patients: 9.6%, controls: 12.9%) (Figure 1B). However, the comparison of the differential presentation of the rs822336 G>C and rs822337 Τ>A genotypes among patients and controls, using the chi-square test, revealed no statistically significant difference (rs822336 G>C, *p* = 0.27; rs822337 T>A, *p* = 0.47).

### 3.2. Association of rs822336 G>C and rs822337 Τ>A with Patients’ Clinical Variables

According to our statistical analysis, both of the studied PD-L1 SNPs demonstrated significant association with TNM stage (rs822336 G>C, *p* = 0.022; rs822337 T>A, *p* = 0.04), DFS (rs822336 G>C, *p* = 0.02; rs822337 T>A, *p* = 0.021) and OS (rs822336 G>C, *p* = 0.04; rs822337 T>A, *p* = 0.004) status. Specifically, regarding the rs822336 G>C, we found that the majority of patients with CG genotype harbored Stage IV breast tumors (81.8%) (Figure 2A). Moreover, this genotype was the most common among the women that recurred (38.9%) and died (66.7%) (Figure 2B and 2C, respectively). Similarly, we found that the rs822337 T>A genotype AA was associated with worse prognosis since it was the most common genotype among stage IV tumors (72.7%) (Figure 3A) as well as in TNBC patients that relapsed (75%) and died (81.5%) (Figure 3B and 3C, respectively).

### 3.3. Survival Analysis and Prognostic Significance of rs822336 G>C and rs822337 Τ>A in TNBC Patients

Survival analysis was performed by Kaplan–Meier analysis and Cox proportional hazards regression models. Disease-free and overall survival information was available for 114 and 108 TNBC patients, respectively, and among them 24 women relapsed (21.1%) and 27 patients died (23.7%). Kaplan–Meier analysis corroborated the abovementioned chi-square test results, since it disclosed that the rs822336 G>C genotype CG was strongly associated with inferior DFS (*p* = 0.002) and OS (*p* = 0.009) intervals compared to the others genotypes (Figure 4).

Univariate and multivariate Cox regression analysis confirmed that the rs822336 G>C genotype CG is a marker of unfavorable prognosis in TNBC, since it was found that it is associated with increased risk of relapse (HR = 4.06, 9% CI = 1.51–4.88, *p* = 0.005) and death (HR = 2.74, 95% CI = 1.18–6.32, *p* = 0.018). Indeed, women harboring the CG allele were 4.06 and 2.74 time more likely to relapse and die, respectively (Table 2).

By performing the same statistical analysis, we found that rs822337 T>A allele AA was associated with worse survival in terms of DFS and OS. Briefly, Kaplan–Meier analysis demonstrated that women with AA genotype are characterized by shorter DFS (*p* = 0.024) and OS (*p* = 0.004) (Figure 5). In concordance with the abovementioned results, Cox regression analysis showed that the rs822337 T>A genotype AA was correlated with high risk of recurrence (HR = 1.02, 95% CI = 0.30–2.45, *p* = 0.04) and death (HR = 4.04, 95% CI = 0.54–3.24, *p* = 0.01) (Table 3).

Based on these results we deemed it interesting to assess the prognostic value of the studied PD-L1 SNPs, after patient categorization based on the combination of the rs822336 G>C and rs822337 Τ>A genotypes (Table 4). The Kaplan-Meier analysis showed that the combination of the CG and AA genotypes are significantly associated with inferior DFS (*p* = 0.002) and OS (*p* = 0.002) intervals (Figure 6). The same conclusion was drawn from the Cox regression analysis, according to which TNBC patients harboring both the CG and AA genotypes demonstrated worse prognosis in terms of DFS (HR = 5.89, 95% CI = 1.35–8.65, *p* = 0.018) and OS (HR = 2.78, 95% CI = 1.02–7.65, *p* = 0.04). Indeed, these women were characterized by almost 6- and 3-times higher risk of relapse and death, respectively.

We then investigated the prognostic performance of the rs822336 G>C and rs822337 Τ>A after the following patients’ categorization: Group I/II/III/VI (Group A) and Group IV/V/VII/VIII (Group B). According to Kaplan–Meier curves, women belonging to group B and harboring mutated genotypes of both SNPs (i.e., CG+AA/CG+AT/CC+AA/CC+AT) had shorter disease-free (*p* = 0.037) (Figure 6C) and overall (*p* = 0.025) (Figure 6D) survival compared to patients having a normal genotype of at least one SNP.

Cox univariate analysis confirmed these results since patients belonging to Group B exhibited inferior DFS compared to those belonging to Group A (HR = 2.45, 95%CI = 1.03–5.98, *p* = 0.04) (Table 5). Taking a step further, we evaluated the independence of the studied SNPs in predicting unfavorable outcomes in TNBC patients by developing a Cox multivariate proportional-hazard regression model adjusted for the combination of the rs822336 G>C and rs822337 Τ>A genotypes, tumor grade, patients’ age, lymph node status and histological type. According to this model, the presence of mutated genotypes of both SNPs was a strong and independent marker of worse prognosis in terms of DFS (HR = 2.89, 95% CI = 1.13–7.87, *p* = 0.02) (Table 5). A similar analysis for the association of the presence of mutated genotypes of both SNPs with patients’ overall survival demonstrated that patients belonging to Group B exhibited shorter OS compared to those belonging to Group A (HR = 2.52, 95% CI = 1.09–5.80, *p* = 0.03) (Table 6). Moreover, the presence of mutated genotypes of both SNPs was a strong and independent marker for predicting shorter OS (HR = 3.44, 95% CI = 1.37–8.61, *p* = 0.008) (Table 6).

## 4. Discussion

The advent of high-throughput genomic technologies inaugurated a new era of molecular genetics, where previously unnoticed genome elements, such as SNPs, represent a new route to assess the etiology of cancer. The possible causative role of SNPs in cancer fueled much interest in recognizing novel cancer biomarkers and therapeutic targets in this class of genetic polymorphisms. Specifically, for TNBC, several studies suggest that SNPs can be used as therapeutic targets and/or prognostic/predictive markers [10,27,28]. As the PD-1/PD-L1 axis has crucial mechanisms as well as a clinical role in TNBC, *PD-1* and *PD-L1* genes merit further investigation for their potential exploitation in the clinical setting of oncology as markers of predicting the 1. risk of TNBC development, 2. course of disease and 3. response to therapy [29].

Considering that mounting evidence indicates that PD-L1 SNPs can be exploited as prognostic markers, we examined the presence of the *PD-L1* SNPs rs822336 G>C and rs822337 T>A in 114 FFPE (formalin-fixed paraffin-embedded) tissues obtained from TNBC patients, and we correlated the SNP status with the clinicopathological and survival data of the patients. The rs822336G>C and rs822337T>A polymorphisms are situated on the promoter region close to the transcription start site. The rs822336C-rs822337A haplotypes are associated with a significant reduction of promoter activity, and it has been found that the rs822336C and rs822337A haplotypes are related with reduced PD-L1 expression at protein level [30]. Both SNPs are located at sites that play a pivotal role in the activation of the promoter, and the region of rs822337T>A represents a significant interaction site between the promoters of *PD-L1* and NF-κΒ [31], whereas it seems that the A allele abolishes the bonding site of the transcription factors SPIB and FOXO3 [32].

These SNPs have been studied for their clinical potential in NSCLC and it has been reported that the co-existence of the rs822336C and rs822337A haplotypes is associated with worse survival, as well as with significantly lower promoter activity and thus with downregulation of *PD-L1* [30]. Specifically, the rs822337T>A polymorphism has been studied as a predictive marker in NSCLC patients after first line paclitaxel-cisplatin chemotherapy, but no statistical significant association was found [33]. A recent study confirmed the abovementioned results regarding the rs822336C polymorphism, since it was found not only that patients with GC or CC alleles demonstrated inferior OS intervals but also that the rs822336C haplotype seems to lead to decreased *PD-L1* expression and/or to the malfunction of the protein [34]. The clinical significance of rs822336G>C has also been investigated in patients with gastric cancer, but the results are inconsistent with those reported for NSCLC as it was found that the rs822336 CC genotype is an independent prognostic marker of favorable prognosis in gastric cancer [35]. This discrepancy is attributed to the different biology of these cancer types.

The first step of our study was the analysis of the differential distribution of the rs822336 G>C and rs822337 T>A polymorphisms among TNBC patients and healthy controls. The statistical analysis demonstrated that both the patients and the healthy controls exhibited the same expressional pattern of the rs822336 G>C and rs822337 T>A polymorphisms. In more detail, we found that not only in patients but also in healthy controls the most abundant allele of the rs822336 G>C was the GG followed by the CG and the CC (*p* = 0.27). Similarly, regarding the rs822337 T>A polymorphism, no significant difference was observed regarding the differential distribution of this polymorphism in the two study cohorts (*p* = 0.47), since in both groups the most abundant genotype was the AA (patients: 53.5%, controls: 46%), followed by the AT (patients: 36.8%, controls: 41.1%) and then the TT genotype (patients: 9.6%, controls: 12.9%).

Next, we aimed to study the association of the rs822336 G>C and rs822337 T>A polymorphisms with important clinicopathological data. By performing the chi square test, we found regarding the rs822336 G>C polymorphism that the majority of patients with the CG genotype harbored Stage IV breast tumors (81.8%), as well as that this genotype was the most common among the women that recurred (38.9%) and died (66.7%). A similar association with markers of worse prognosis was observed for the rs822337 T>A polymorphism as well. More specifically, it was found that AA was the most common genotype among stage IV tumors (72.7%) as well as in TNBC patients that relapsed (75%) and died (81.5%). Overall, these results underline that the rs822336C and rs822337A haplotypes, which are associated with decreased expression levels of PD-L1, are significantly correlated with markers of unfavorable prognosis. This observation can be interpreted based on the known data regarding the mechanistic and clinical role of these SNPs. In more detail, as mentioned above, Lee et al. stated that rs822336C and rs822337A haplotypes are related with reduced PD-L1 expression [30], whereas at the same time it has been found that breast cancer patients with decreased PD-L1 expression are characterized by worse survival rates [36] as well as that in TNBC the PD-L1 downregulation is associated with unfavorable prognosis [37]. Hence, by combining these data we can conclude that the rs822336C and rs822337A haplotypes can act as indicators of unfavorable prognosis since they can lead to reduced expression of the PD-L1. 

The final step of our analysis was the performance of a thorough survival analysis in order to examine the association of the SNPs’ distribution with disease-free (DFS) and the overall survival (OS). Both Kaplan–Meier and Cox regression analysis demonstrated that the rs822336 G>C genotype CG is strongly associated with inferior DFS (Kaplan–Meier analysis *p* = 0.002, Cox analysis, *p* = 0.005) and OS (Kaplan–Meier analysis; *p* = 0.009, Cox analysis; *p* = 0.018) intervals compared to the others genotypes (Figure 4). Moreover, a univariate Cox analysis revealed that women harboring the CG allele were 4.06 and 2.74 time more likely to relapse and die, respectively. The same statistical analysis revealed that rs822337 T>A allele AA is associated with worse survival in terms of DFS and OS. According to Kaplan–Meier analysis, patients with the AA genotype are characterized by shorter DFS (*p* = 0.024) and OS (*p* = 0.004). Cox regression analysis confirmed these results; it was found that rs822337 T>A genotype AA is marginally correlated with high risk of recurrence (*p* = 0.04) but most importantly with a four-fold higher risk of death (*p* = 0.01).

Taking a step further, we examined the prognostic significance of the rs822336 G>C and rs822337 T>A polymorphisms after patients’ categorization according to Table 6. Kaplan–Meier and Cox regression analyses showed that the combination of the CG and AA genotypes is significantly associated with inferior DFS (Kaplan–Meier; *p* = 0.002, Cox analysis; *p* = 0.018) and OS (Kaplan–Meier; *p* = 0.002, Cox analysis; *p* = 0.04) intervals. Moreover, Kaplan–Meier and univariate Cox analysis revealed that women harboring mutated genotypes of both SNPs (i.e., CG+AA/CG+AT/CC+AA/CC+AT) had shorter disease-free (Kaplan–Meier; *p* = 0.037, Cox analysis; *p* = 0.04) and overall (Kaplan–Meier; *p* = 0.025, Cox analysis; *p* = 0.03) survival, compared to patients having normal genotypes of at least one SNP. Multivariate analysis adjusted for significant clinipathological features (i.e., tumor grade, patients age, lymph node status and histological type) also showed that the presence of mutated genotypes of both SNPs is a strong and independent marker for predicting shorter DFS (HR = 2.89, 95% CI = 1.13–7.87, *p* = 0.02) and OS (HR = 3.44, 95% CI = 1.37–8.61, *p* = 0.008). Overall, our statistical analysis reinforced the abovementioned significant association of the rs822336C and rs822337A haplotypes with unfavorable prognosis, since it was found that these haplotypes (individually or in combination) are markers of shorter DFS and OS. Our results are in line with a previous study, according to which NSCLC patients harboring the rs822336CC and rs822337AA gonotypes demonstrated a marginally significant decreased OS, whereas the same group concluded that the rs822336C and rs822337A haplotypes were associated with inferior overall survival intervals [30].

A noteworthy biological implication of our approach is the interplay between polymorphisms and the glycosylation patterns of PD-L1. Studies have shown that TNBC cells have higher levels of glycosylated PD-L1 [38]. This feature affects the quality of immunohistochemical (IHC) staining procedures, generating more false negative results, and potentially excluding from therapy tumors that would likely be responsive to anti-PD-L1 therapies [39,40]. In fact, the underestimation of PD-L1 expression in breast tumor tissues has been suggested as a potential underlying cause for the reduced benefit of atezolizumab plus nab-paclitaxel in the post-market phase-III trial that led to its withdrawal from the indication in TNBC in 2021 [38]. Specifically, in the IMpassion130 study, patients were stratified using standard IHC methods, whereas a preceding de-glycosylation process of breast tissue samples has been proposed as a more accurate approach to assess PD-L1 expression and therefore atezolizumab effectiveness [24]. Interestingly, N-glycosylation occurs after the transfer of a glycan to an asparagine (Asn) amino acid side-chain acceptor [41]. Since Asn is coded by synonymous codons AAC and AAT, the studied rs822337 AT genotype may affect glycosylation (and concurrently response) to anti-PD-L1 therapy. Since AT was one of the most abundant PD-L1 genotypes in both groups included in our study, these results are worthy of attention and can serve as a basis for further investigation on the relationship between the level of PD-L1 glycosylation and response to anti-PD-L1 therapy.

Our study represents a first attempt to investigate the clinical impact in TNBC of two known *PD-L1* polymorphisms which have attracted much interest recently, due to their implication in expressional regulation of PD-L1, a molecule that according to mounting evidence is in the core not only of certain cancer-related events but also of clinical management of several cancer types, including TNBC. According to our data the rs822336C and rs822337A haplotypes are associated with markers of unfavorable prognosis in TNBC patients as well as with inferior DFS and OS intervals. Moreover, the presence of mutated genotypes of both SNPs is a strong and independent marker for predicting shorter DFS and OS.

## 5. Conclusions

In conclusion, PD-L1 rs822336 G>C and rs822337 T>A polymorphisms are differentially expressed in TNBCs, and this distribution is associated with markers of unfavorable prognosis and with worse patient survival. Considering how polymorphisms exert not only transcriptional but also a wide range of post-translational effects, including affecting the level of *PD-L1* glycosylation, our findings may have biological and clinical implications in the elucidation of TNBC’s aggressive phenotype. Further well-designed investigations in a larger cohort of patients should be performed to elucidate the underlying mechanisms by which these SNPs affect the prognosis of TNBC patients, and to verify our results.

## Figures and Tables

**Figure 1 medicina-58-01399-f001:**
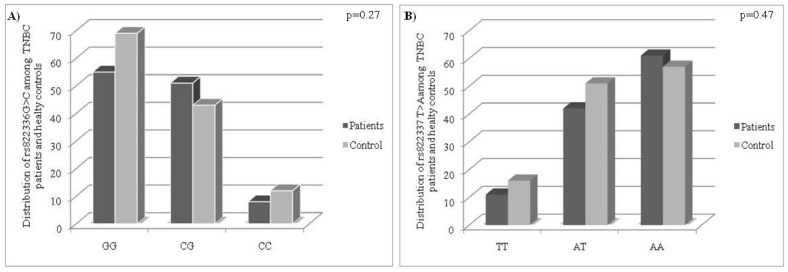
Distribution of the GG/CG/CC (rs822336 G>C) (**A**) and AA/AT/TT (rs822337 Τ>A) (**B**) alleles among TNBC patients and healthy controls.

**Figure 2 medicina-58-01399-f002:**
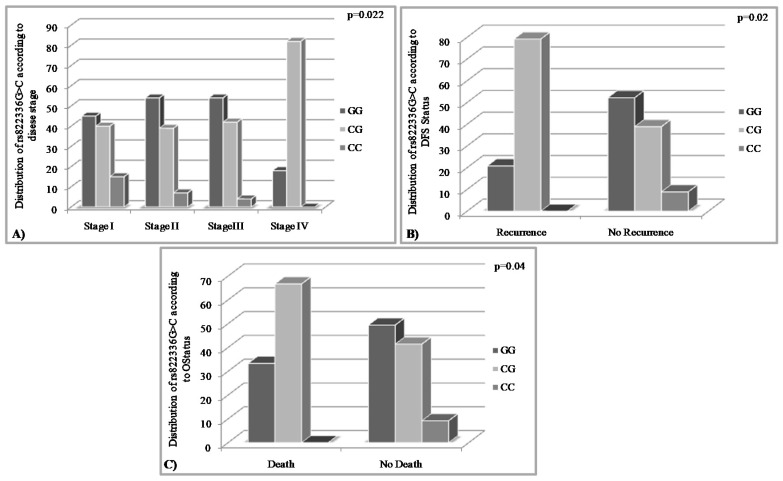
Distribution of the rs822336 G>C according to TNM Stage (**A**), DFS (**B**) and OS Status (**C**).

**Figure 3 medicina-58-01399-f003:**
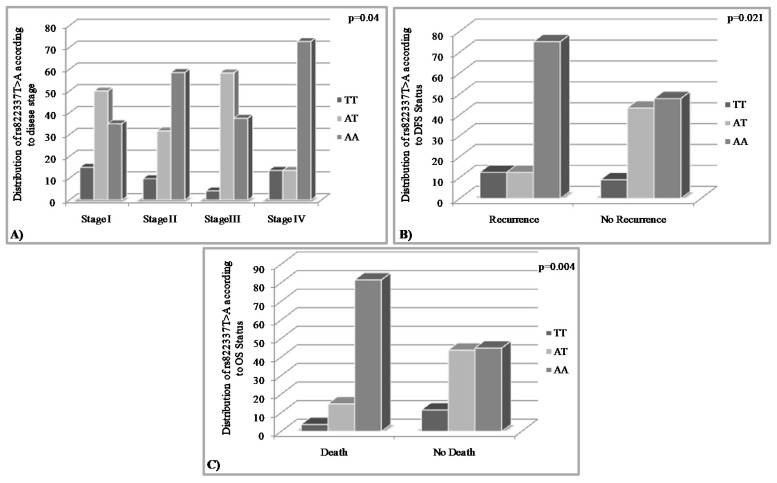
Distribution of the rs822337 T>A according to TNM Stage (**A**), DFS (**B**) and OS Status (**C**).

**Figure 4 medicina-58-01399-f004:**
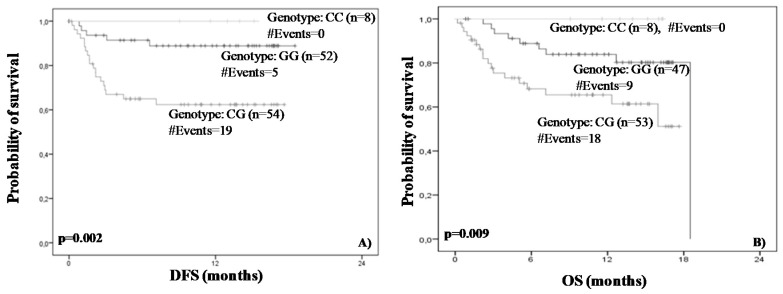
Kaplan–Meier disease-free (**A**) and overall (**B**) survival curves. *p* values were calculated by log-rank test.

**Figure 5 medicina-58-01399-f005:**
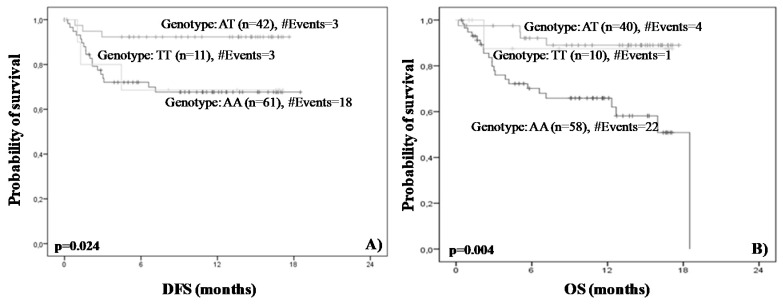
Kaplan–Meier disease-free (**A**) and overall (**B**) survival curves. *p* values were calculated by log-rank test.

**Figure 6 medicina-58-01399-f006:**
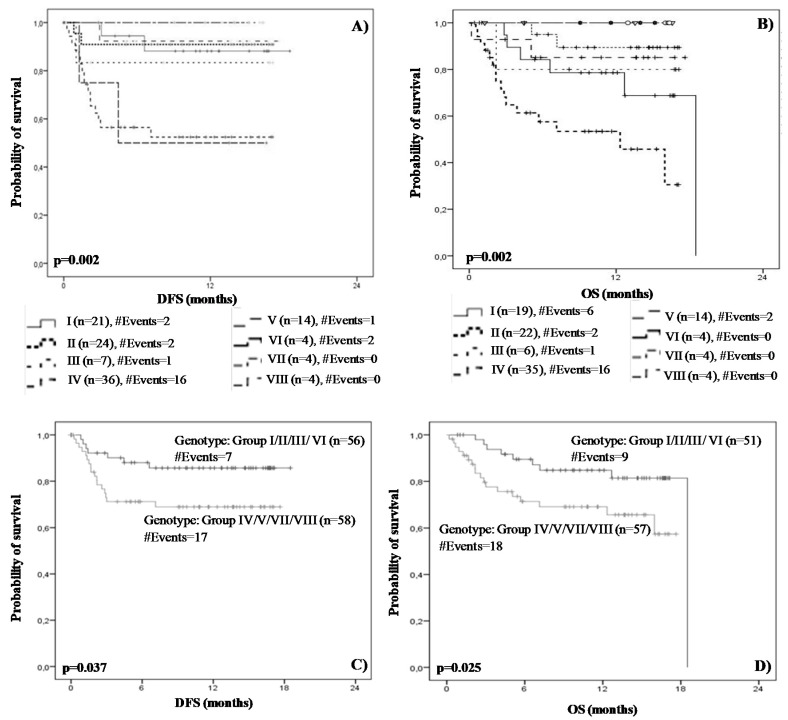
Kaplan–Meier disease-free (**A**,**C**) and overall (**B**,**D**) survival curves, after patients’ stratification. *p* values were calculated by log-rank test.

**Table 1 medicina-58-01399-t001:** Clinicopathological characteristics of triple-negative breast cancer patients (*n* = 114).

Variant	Number of Patients	Variant	Number of Patients
Age		Ki-67 Index	
≥57	57 (50.0%)	Positive	58 (50.9%)
<57	56 (49.1%)	Negative	15 (13.1%)
Unknown	1(0.9%)	Unknown	41 (36.0%)
ΤΝΜ Stage		Grade	
I	20 (17.5%)	I	2 (1.7%)
II	41 (35.9%)	II	10 (8.8%)
III	24 (21.1%)	III	95 (83.4%)
IV	22 (19.4%)	Unknown	7 (6.1%)
Unknown	7 (6.1%)		
Tumor Type		Lymph nodes	
Ductal	87 (76.3%)	N0	66 (57.9%)
Lobular	10 (8.8%)	N1	13 (11.4%)
Medular	8 (7%)	N2	9 (7.9%)
Other	9 (7.9%)	N3	19 (16.7%)
Unknown	0 (0%)	Unknown	7 (6.1%)
Disease Progression		Death	
Yes	24 (21.1%)	Yes	21 (18.4%)
No	90 (78.9%)	No	93 (81.6%)
Unknown	0 (0%)	Unknown	0 (0%)

**Table 2 medicina-58-01399-t002:** Cox univariate regression analysis of rs822336 G>C for the prediction of disease-free (DFS) and overall survival (OS).

Disease-Free Survival (DFS) (*n* = 114)			
**Variable**	**HR ^a^**	**95% CI ^b^**	***p* Value**
*rs822336*			
GG	1.00		0.005
CG	4.06	1.51–4.88	
CC	0.00	0.00	
**Overall Survival (OS) (*n* = 108)**			
**Variable**	**HR ^a^**	**95% CI ^b^**	***p* Value**
*rs822336*			
GG	1.00		0.018
CG	2.74	1.18–6.32	
CC	0.00	0.00	

^a^ Hazard Ratio, ^b^ Confidence interval of the estimated HR. Bold value indicates statistical significance

**Table 3 medicina-58-01399-t003:** Cox univariate regression analysis of rs822337 T>A for the prediction of disease-free (DFS) and Overall survival (OS).

Disease-Free Survival (DFS) (*n* = 107)			
**Variable**	**HR ^a^**	**95% CI ^b^**	***p* Value**
*rs822337*			
TT	1.00		0.04
AT	0.22	0.04–1.08	
AA	1.02	0.30–2.45	
**Overall Survival (OS) (*n* = 108)**			
**Variable**	**HR ^a^**	**95% CI ^b^**	***p* Value**
*rs822337*			
TT	1.00		0.01
AT	0.87	0.10–2.75	
AA	4.04	0.54–3.24	

^a^ Hazard Ratio, ^b^ Confidence interval of the estimated HR. Bold value indicates statistical significance

**Table 4 medicina-58-01399-t004:** Patient stratification based on the combination of the rs822336 G>C and rs822337 Τ>A genotypes.

Group	rs822336 Genotype	rs822337 Genotype	Total	Recurrence	Death
Ι	GG	AA	21	2 (9.5%)	6 (28.6%)
ΙΙ	GG	AT	24	2 (8.3%)	2 (8.3%)
ΙΙΙ	GG	TT	7	1 (14.3%)	1 (14.3%)
ΙV	CG	AA	36	16 (44.4%)	16 (44.4%)
V	CG	AT	14	1 (7.1%)	2 (14.2%)
VI	CG	TT	4	2 (50%)	0
VII	CC	AA	4	0	0
VIII	CC	AT	4	0	0
IX	CC	TT	-	-	-

**Table 5 medicina-58-01399-t005:** Cox univariate and multivariate regression analysis for the prediction of disease-free survival (DFS) after stratification of study cohort according to combination of the rs822336 G>C and rs822337 Τ>A genotypes.

Univariate Analysis (*n* = 107)			
**Variable**	**HR ^a^**	**95% CI ^b^**	***p* Value**
*Group*			
A	1.00		0.04
Β	2.45	1.03–5.98	
**Multivariate Analysis (*n* = 98)**			
**Variable**	**HR ^a^**	**95% CI ^b^**	***p* Value**
*Group*			
A	1.00		0.02
Β	2.89	1.13–7.87	
Grade	1.37	0.37–5.11	0.63
Age	0.63	0.26–1.50	0.29
Lymph node status	1.35	0.96–1.90	0.07
Histology	1.15	0.75–1.76	0.52

^a^ Hazard Ratio, ^b^ Confidence interval of the estimated HR. Bold value indicates statistical significance.

**Table 6 medicina-58-01399-t006:** Cox univariate and multivariate regression analysis for the prediction of overall survival (OS) after stratification of study cohort according to combination of the rs822336 G>C and rs822337 Τ>A genotypes.

Univariate Analysis (*n* = 108)			
**Variable**	**HR ^a^**	**95% CI ^b^**	***p* Value**
*Group*			
A	1.00		0.03
Β	2.52	1.09–5.80	
**Multivariate Analysis (*n* = 98)**			
**Variable**	**HR ^a^**	**95% CI ^b^**	***p* Value**
*Group*			
A	1.00		0.008
Β	3.44	1.37–8.61	
Grade	0.94	0.28–3.10	0.92
Age	1.88	0.79–4.45	0.15
Lymph node status	1.63	1.19–2.22	0.002
Histology	0.86	0.55–1.34	0.51

^a^ Hazard Ratio, ^b^ Confidence interval of the estimated HR. Bold value indicates statistical significance.

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
