# Peer review of "PD-L1 Gene Polymorphisms rs822336 G>C and rs822337 T>A: Promising Prognostic Markers in Triple Negative Breast Cancer Patients"

_medicina, 2022, doi:10.3390/medicina58101399_

Round 1

Reviewer 1 Report

The manuscript written by Makrantonakis et al. investigates the prognostic role of the rs822336 G>C and rs822337 T>A polymorphism of the PD-L1 in triple-negative breast cancer patients.

Regardless of the fact that it is an interesting work, some changes should be made to the manuscript.

In the section Materials and Methods:

1.     In the paper, the authors refer to table 1, which is not found anywhere in the paper.

In the section Results:

2.     The results in table 2 and figure 1 are identical. It is necessary to delete one.

3.     The results in table 3 and figure 2 are identical. It is necessary to delete one.

4.     The results are very difficult to follow. Also, it is necessary to compare some additional clinical and biochemical data of patients for some serious conclusions.

Author Response

The manuscript written by Makrantonakis et al. investigates the prognostic role of the rs822336 G>C and rs822337 T>A polymorphism of the PD-L1 in triple-negative breast cancer patients.

Regardless of the fact that it is an interesting work, some changes should be made to the manuscript.

We thank the reviewer for thoroughly reading our work and for their time and effort in reviewing our manuscript.

In the section Materials and Methods:

  1. In the paper, the authors refer to table 1, which is not found anywhere in the paper.

Thank you for pointing this out. We rectify this important omission in the revised version of our manuscript by presenting the Clinicopathological Characteristics of triple-negative breast cancer patients included in our study in Table 1.

In the section Results:

  1. The results in table 2 and figure 1 are identical. It is necessary to delete one.

Thanks to your valuable comment, only Figure 1 is now included in the revised version of our manuscript.

  1. The results in table 3 and figure 2 are identical. It is necessary to delete one.

Thank you for your observation, we removed Table 3 from the revised manuscript according to your helpful comment.

  1. The results are very difficult to follow. Also, it is necessary to compare some additional clinical and biochemical data of patients for some serious conclusions.

We have revised the Result section and divided it into sub-sections to make it easier to follow. We also added a paragraph in the Discussion Section that presents potential clinical implications of our findings, in relation to their biologic background and attempted to draw some more noteworthy conclusions.

Reviewer 2 Report

Dear Editor in Chief

I found the manuscript should make some minor improvement in the style of the abstract.

Author Response

Thank you for your time and efforts in reviewing our work. We revised the abstract of our manuscript to make it more reader friendly and we hope it now meets with your approval.

Reviewer 3 Report

In this study authors aimed to investigate the prognostic role of the rs822336 G>C and rs822337 T>A polymorphism of the PD-L1 in TNBC patients.

I have the following concerns regarding the main concepts and probable impact of this research in the current drug development, especially anti-PD-L1 therapies.

Introduction:

1- Unfortunately, the TNBC indication of atezolizumab was withdrawn in August 2021 due to a failure of a post-market Phase III clinical trial (reference: PMID: 35530278). Therefore, authors should add it in the Introduction when they are talking about atezolizumab; however, pembrolizumab (Keytruda®), a monoclonal antibody against PD-1, was approved for treatment of TNBC patients in July 2021 (reference: PMID: 35530278). Hence, the background should be updated based on the recent advances in the field.

Discussion:

2- Recent study has also shown the lack of response to anti-PD-L1 therapy (comment 1) may be because of high level of glycosylation of TNBC patients, therefore those patients with lower glycosylation will respond to atezolizumab (reference: PMID: 35141008). As the glycosylation will be done using Asn amino acid residue whose codons are AAT and AAC, PD-L1 polymorphism at AT may affect glycosylation, and therefore, response to anti-PD-L1 therapy. It would be interesting, if authors discuss this challenge in the current anti-PD-L1 therapy. In my opinion, (MAYBE) AT genotype has higher glycosylation with lower response to PD-L1 therapy. and if the genotype of patient determined before administration, we can manage the disease better. It was interesting, if I saw the level of glycosylation of collected samples in your study. It is doable by histochemistry.

Conclusion:

Anyway, please think about it and let your reader know about it. In my opinion your polymorphism data can predict the level of glycosylation in PD-L1 and it is interesting for our future anti-PD-L1 drug development. You can also highlight it in the Conclusion.

Figures:

3- Figure 1: Data presented in Figure 1 are not significant. Authors made it confusing when mixed the caption with data presented in other Figures and Tables.

Please, explain each Figure and Table in its own caption. You can then discuss it in the text. Please, do not discuss them in Figure caption.

4- Where is Table 1? Authors started with Table 2. Please, check the numbers of Tables and cite them in the text probably.

References:

5- Many authors cited without their surnames. Please, carefully check the authors' names in the references. Such as Ref.1,3-8, 10-17, 19-21, and 33-34.

Author Response

1- Unfortunately, the TNBC indication of atezolizumab was withdrawn in August 2021 due to a failure of a post-market Phase III clinical trial (reference: PMID: 35530278). Therefore, authors should add it in the Introduction when they are talking about atezolizumab; however, pembrolizumab (Keytruda®), a monoclonal antibody against PD-1, was approved for treatment of TNBC patients in July 2021 (reference: PMID: 35530278). Hence, the background should be updated based on the recent advances in the field.

We thank the reviewer for pointing this out. Clarifications have been added to the main manuscript under the Introduction section (Lines 72-74 and 91-93) that more accurately reflect the current literature.

Discussion:

2- Recent study has also shown the lack of response to anti-PD-L1 therapy (comment 1) may be because of high level of glycosylation of TNBC patients, therefore those patients with lower glycosylation will respond to atezolizumab (reference: PMID: 35141008). As the glycosylation will be done using Asn amino acid residue whose codons are AAT and AAC, PD-L1 polymorphism at AT may affect glycosylation, and therefore, response to anti-PD-L1 therapy. It would be interesting, if authors discuss this challenge in the current anti-PD-L1 therapy. In my opinion, (MAYBE) AT genotype has higher glycosylation with lower response to PD-L1 therapy. and if the genotype of patient determined before administration, we can manage the disease better. It was interesting, if I saw the level of glycosylation of collected samples in your study. It is doable by histochemistry.

We thank the reviewer for their well-researched and noteworthy viewpoint. Extensive literature presents the interplay among the levels of glycosylated PD-L1 and responsiveness of TNBC patients to atezolizumab. In this context, and taking into consideration your thoughtful comment, we made an effort to present in Lines 362-377the aforementioned observations, which is experimentally beyond the scope of the current study but would be an interesting future endeavor, as we point out in the revised text.

Conclusion:

Anyway, please think about it and let your reader know about it. In my opinion your polymorphism data can predict the level of glycosylation in PD-L1 and it is interesting for our future anti-PD-L1 drug development. You can also highlight it in the Conclusion.

Once again, we appreciate your constructive input. We have highlighted in the conclusion (Lines 389-371) the potential usage of our initial polymorphism data as a basis for predicting PD-L1 glycosylation level.

Figures:

3- Figure 1: Data presented in Figure 1 are not significant. Authors made it confusing when mixed the caption with data presented in other Figures and Tables.

Please, explain each Figure and Table in its own caption. You can then discuss it in the text. Please, do not discuss them in Figure caption.

Thank you for your comments. Please find the requested changes in the revised manuscript.

4- Where is Table 1? Authors started with Table 2. Please, check the numbers of Tables and cite them in the text probably.

We rectify this important omission in the revised version of our manuscript by presenting Clinicopathological characteristics of triple-negative breast cancer patients included in our study in Table 1, We also re-numbered all the included Tables.

References:

5- Many authors cited without their surnames. Please, carefully check the authors' names in the references. Such as Ref.1,3-8, 10-17, 19-21, and 33-34.

As you suggested, we corrected the references' format according to the journal guidelines.